# Influence of Extruded Tubing and Foam-Filler Material Pairing on the Energy Absorption of Composite AA6061/PVC Structures

**DOI:** 10.3390/ma16186282

**Published:** 2023-09-19

**Authors:** John Magliaro, Pouya Mohammadkhani, Foad Rahimidehgolan, William Altenhof, Ahmet T. Alpas

**Affiliations:** Department of Mechanical, Automotive and Materials Engineering, University of Windsor, 401 Sunset Avenue, Windsor, ON N9B 3P4, Canada; jmagliar@uwindsor.ca (J.M.); mohamm6a@uwindsor.ca (P.M.); rahimid@uwindsor.ca (F.R.); aalpas@uwindsor.ca (A.T.A.)

**Keywords:** energy absorption, composite structures, lightweight materials, AA6061, PVC foam, axial cutting, axial crushing

## Abstract

There is accelerating demand for energy-absorbing structures fabricated from lightweight materials with idealized, near-constant force responses to simultaneously resolve the engineering challenges of vehicle mass reduction and improved occupant safety. A novel compounded energy dissipation system composed of AA6061-T6 and AA6061-T4 tubing subjected to hybrid cutting/clamping and H130, H200 and H250 PVC foam compression was investigated utilizing quasi-static experiments, finite element simulations and theoretical modeling. Identical structures were also subjected to axial crushing to compare with the current state of the art. The novel cutting/foam crushing system exhibited highly stable collapse mechanisms that were uniquely insensitive to the tube/foam material configuration, despite the disparate material properties, and exceeded the energy-absorbing capacity and compressive force efficiency of the axial crushing mode by 14% and 44%, respectively. The simulated deformation profiles and force responses were consistent with the experiments and were predicted with an average error of 12.4%. The validated analytical models identified numerous geometric/material configurations with superior performance for the compounded AA6061/PVC foam cutting/foam crushing system compared to axial crushing. An Ashby plot comparing the newly obtained results to several findings from the open literature highlighted the potential for the compounded cutting/foam crushing system to significantly outperform several alternative lightweight safety systems.

## 1. Introduction

The field of structural crashworthiness has experienced considerable growth both in terms of the community size and extent of contributions since its inception in the 20th century [1,2,3]. Large-capacity, sacrificial energy absorbers that collapse along their axes, commonly referred to as front and rear rails or crash boxes, are integrated within the frames of all modern vehicles, representing the first line of defense during accidents, collisions and derailments. A multitude of innovations to this technology have been proposed to address the ever-growing challenge of improved occupant safety, including structural discontinuities [4,5,6], multi-celled profiles [7,8], origami-type initiators [9,10,11], the advent of composite materials [12,13,14,15,16,17] and bio-inspired solutions [18,19,20], among many others. Accelerating demand for electric road vehicles and high-speed rail systems composed of lightweight materials, with the overarching goals of sustainable manufacturing and consumption practices, has posed many new engineering challenges. However, this ongoing transformation also offers a rare opportunity to reset and adopt many novel technologies to advance the transportation sector, including lightweight safety systems.

Crash boxes filled with structural foam cores represent one of the most thoroughly studied enhancements for the universally implemented axial crushing energy dissipation mode [2,21,22,23]. Foam-filled composite structures generally experience increased energy-absorbing capacity and collapsed profile stability during oblique loading compared to their hollow counterparts [24,25,26] at the expense of reduced mass-specific energy absorption and increased cost [27,28,29]. The inclusion of foam cores within axially crushed tubing has also been, to date, ineffective in eliminating the intensely fluctuating local minima and maxima (secondary peaks and valleys) inherent to the axial crushing force response [30,31,32,33]. This phenomenon has limited the effectiveness of the axial crushing energy dissipation mode, for thin-walled structures with and without foam cores, since these peaks and valleys in the force response deviate significantly from the intended mean crushing force corridors, and thus tend to contribute toward occupant deaths and injuries [34].

Hybrid energy absorbers, which subject the same thin-walled structures to more ideal cutting, splitting and tearing-based deformation modes [21,35,36], have recently emerged as promising, higher-capacity alternatives to axial crushing which can eliminate the dangerous and notoriously challenging fluctuations in the force response associated with this technology. Notable examples include combined metal/CFRP structures [30,37,38,39], compounded axial splitting/expansion [40,41,42], expansion/shrinking [43], splitting/shrinking [44,45] and axial cutting/radial clamping [46,47]. The latter deformation mode can meet and/or exceed the energy-absorbing capacity of the traditional axial crushing mode while simultaneously improving the force efficiency (i.e., reducing reaction force fluctuations) by a factor of 2 for sacrificial AA6061 energy-absorbing structures [48].

The concept of introducing structural foam cores to further enhance the mechanical performance of these novel alternative deformation modes has received some attention in recent years [49,50]. Near-constant reaction forces were experimentally achieved for extruded metallic tubing subjected to axial splitting with simultaneously occurring uniform compression of solid [51] and sandwiched [52,53] aluminum-based Alulight^®^ (Alulight GmbH, Ranshofen, Austria) internal foam cores. Closed-cell polyvinyl chloride (PVC) foams have also shown promise for enhancing many lightweight structural applications, including various energy absorbers [54,55,56], and were correspondingly combined with AA6061 tubing subjected to axial cutting [57] and hybrid cutting/clamping [58] with high efficacy. However, the concept is relatively new, and thus findings in the current literature are limited. Most published investigations to date have emphasized studying foam cores with relative densities below 10%; higher-density structural foams must be considered for high-capacity systems. Considering the influence of extruded material conditions is another underserviced but necessary consideration since lightweight energy-absorbing materials (aluminum alloys, for example) are highly susceptible to brittle fracture during axial collapse [2,3,18].

The present study contains comprehensive experimental, numerical and analytical investigations on the influence of extruded material annealing and foam core density on the energy-absorbing capacity, efficacy and limitations for AA6061/H-series PVC foam-filled energy-absorbing structures subjected to axial crushing and 10-bladed cutting/clamping. Extruded AA6061 tubing, in both T6 (as-received) and T4 (annealed) temper conditions, was considered to assess the extent and severity of brittle material fracture for PVC foam cores with relative densities between 0.092 and 0.171. Experiments and numerical modeling were performed for the hollow and PVC foam-filled structures subjected to compounded, 10-bladed cutting/foam crushing. Complementary axial crushing tests and simulations allowed for direct comparisons between the newly proposed energy absorber and the current state of the art. The novel compounded axial cutting/foam crushing mode displayed significant benefits over the traditional axial crushing mode, especially with respect to energy-absorbing capacity and stability during collapse, for every considered tube/foam core pair. The novel compounded cutting/foam crushing remained uniquely stable during collapse regardless of the disparate material pairings. The numerical models accurately replicated the experimentally observed force/displacement responses and deformation profiles. A theoretical study model was utilized to newly identify a wider range of material pairings and critical structural dimensions which benefited most significantly from the compounded cutting/foam crushing mode.

## 2. Materials and Methods

The following section provides details on the composite AA6061/PCV foam energy-absorbing structures, including the fundamental material properties, critical details for each constitutive component and fixture, experimental testing and numerical modeling procedures and the performance metrics necessary to analyze the data. A parametric scope emphasizing the influence of the tubing/foam core material pair is also presented.

### 2.1. Material Preparation

#### 2.1.1. AA6061 Extruded Tubing

Seamless AA6061-T6 tubing (Essex Metals, 1905 Blackacre Drive, Oldcastle, ON, Canada) with a 63.5 mm outer diameter and 1.588 mm wall thickness was (rough) saw-cut into specimens with nominal lengths of 200 mm, followed by end milling to remove chips and burrs. This geometry was selected based on nominal material sizes/availability and because the tube ratio, $\frac{d_{o}}{2t}$, far exceeded an order of magnitude (i.e., they were thin-walled), a prerequisite for lightweight safety system design [18,21,36]. The equivalent flow stresses, *σ_o_*, utilized in the theoretical studies presented in Section 4.5, were calculated as
(1)$$\sigma_{o}=\frac{1}{\varepsilon_{f}-\varepsilon_{Y}}\int_{\varepsilon_{Y}}^{\varepsilon_{f}}\sigma_{p}\left(\varepsilon \right)d\varepsilon$$
where *ε_Y_* and *ε_f_* represent the strains to yielding and fracture, respectively, and $\sigma_{p}\left(\varepsilon \right)$ represents the plastic regime of each respective stress/strain response.

The parametric scope considered extruded tubing in the mentioned T6 temper condition and an annealed T4 temper condition with a lower flow strength but significantly improved strain-to-failure. The latter specimens were obtained by heat-treating a portion of the as-received (T6 temper condition) samples per ASM Handbook guidelines [59]. Characteristic engineering stress/strain responses annotated with the flow stress of the AA6061 material in each temper condition are provided in Figure 1. The differences in plastic behavior between the T6 and T4 temper conditions were associated with differences in their respective microstructures, with the former primarily characterized by smaller average grain sizes than the latter. The larger and less evenly distributed grains of the T4 temper condition form during the natural aging process associated with the heat treatment which serves as an annealing process, eliminating residual stresses from the extruding process and correspondingly reducing the overall strength [60,61]. The annealing process also decomposes the hardened precipitates from the T6 temper condition into solute atoms, further reducing the yield strength but dramatically improving the ductility.

#### 2.1.2. H-Series PVC Foam Cores

The internal foam cores were composed of H-series, closed-cell PVC foam in H130, H200 and H250 configurations (Diab Group, 315 Seahawk Drive, DeSoto, TX, USA) with structural densities of 126.7 kg/m^3^, 181.2 kg/m^3^ and 235.6 kg/m^3^, respectively. Representative quasi-static compressive stress/strain responses for each material are provided in Figure 2a,b for the rise/out-of-plane (RD) and in-plane (IPD) directions, respectively; note that the foam cores were stacked (and crushed) parallel to the rise direction in the apparatus discussed in Section 2.2. The stress/strain responses of the H-series PVC foams, $\sigma_{f}\left(\varepsilon \right)$, were replicated utilizing the Avalle constitutive model [62]:(2)$$\sigma_{f}\left(\varepsilon \right)=K_{p}\cdot \left(1-e^{-\frac{E}{K_{p}}\varepsilon {\left(1-\varepsilon \right)}^{m}}\right)+K_{h}{\cdot \left(\frac{\varepsilon }{1-\varepsilon }\right)}^{n}$$

The necessary material coefficients, obtained via regression analysis, and critical mechanical properties are summarized in Table 1 for the rise (out-of-plane) and in-plane directions, respectively.

#### 2.1.3. Composite AA6061/PVC Foam Structures

The AA6061 tubes were combined concentrically with internal PVC foam cores, the latter were obtained from 25.4 mm-thick rectangular sheets utilizing a hole saw to match the inner profile of the extruded tubing. The cylindrical foam cores were fabricated utilizing a hole saw mounted to a Bridgeport^®^ with the arbor removed from rectangular sheets, consistent with previous studies [56,57,64]. A 0.40 mm clearance fit was achieved between these entities, and no adhesives were applied at the tubing/foam contact interface.

### 2.2. Energy-Absorbing Apparatus

The sacrificial AA6061/PVC foam composite structures were subjected to a traditional axial crushing deformation mode and a novel, in-house designed, engineered and fabricated 10-bladed hybrid cutting/clamping mode, shown in Figure 3a and Figure 3b, respectively. The 10-bladed cutter (Figure 3c) was paired with a conical deflector (Figure 3d) to provide a standoff. An internal platen was positioned concentrically within the tubing profile, above the cutter and below the foam cores, to apply uniform compression of these entities while the extruded material was simultaneously subjected to cutting. Both individual deformation modes, namely the steady-state cutting and plateau compression of the foam, promote idealized force responses [56], yielding a combined high-capacity energy dissipation mechanism.

### 2.3. Experimental Approach

#### 2.3.1. Testing Procedure

Experimental testing was completed utilizing apparatuses consistent with the schematics from Figure 3a,b for the axial crushing and cutting deformation modes, respectively, contained within a 600 kN-capacity universal test frame (MTS Systems, 14,000 Technology Drive, Eden Prairie, MN, USA). The apparatuses were instrumented with a 222 kN PCB^®^ 1204-13A load cell (PCB Piezotronics, 3425 Walden Avenue, Depew, NY, USA) and 300 mm-range AR700-12 Acuity^®^ non-contact laser displacement transducer (Schmitt Measurement Systems, 2765 NW Nicolai Street, Portland, OR, USA). All measurements were synchronized using an NI CompactDAQ system (cDAQ-9174) operating at 2 kHz (National Instruments, 11,500 N Mopac Expwy, Austin, TX, USA). The test frame was operated at 10 mm/min, limiting the nominal compressive strain rate within the foam cores to 10^−3^ s^−1^, for 125 mm of displacement (62.5% and 66% axial compression for the extruded tubing and foam cores, respectively).

#### 2.3.2. Parametric Scope

The scope of experimental testing was selected to characterize the influence of the tube/foam core material pairing on the energy-absorbing capacity, effectiveness and structural stability of the traditional axial crushing (AC) and novel 10-bladed hybrid cutting/clamping (C10) deformation modes. The results were identified with the naming convention: EM/FC- DM-#, where EM/FC indicated the tubing material/foam core pairing, DM the deformation mode and # the individual test number or type of model (and ‘Num’ for numerical). The full parametric scope is summarized in Table 2; two tests were completed per specimen group for repeatability.

### 2.4. Finite Element Modeling

#### 2.4.1. Mesh Properties and Discretization

Representative finite element meshes for the axial crushing and compounded cutting/foam crushing systems are provided in Figure 4, respectively. The axial crushing mode was modeled with fully integrated shell elements for the AA606 tubing, with a characteristic element length of 1.588 mm. A solid material-and-void Eulerian element formulation, with an accompanying airmesh (control volume), was utilized to model the cutting deformation mode with 3 elements through the wall thickness. The airmesh was truncated at the boundary of the cutting tool (i.e., only the portion that interacts with the cutter was considered) to mitigate computation requirements in accordance with previously validated studies [45,56]. Solid elements with a 3 mm characteristic edge length and fully integrated formulation were utilized for the PVC foam cores. A translational spring was included between the foam cores for the 10-bladed cutting/clamping deformation mode to account for in-plane volumetric expansion of higher-density PVC foams (i.e., relative densities exceeding 0.10), discussed further in Section 2.4.5. The symmetry conditions visible in Figure 4b are outlined in Section 2.4.6.

#### 2.4.2. AA6061 Extruded Tubing Material Models

The AA6061 tubing was modeled utilizing a simplified Johnson–Cook relationship as defined in Equation (3), since the present study considered quasi-static loading where thermal effects such as thermal softening are negligible. To that end, a simple linear polynomial equation of state with the bulk modulus, *K*, defined as the coefficient of proportionality was implemented. The material parameters and exponential constants for each temper condition are summarized in Table 3.
(3)$$\sigma_{Y}=\left(A+B{\bar{\varepsilon }}^{p^{n}}\right)\left(1+c\ln\;{\dot{\varepsilon }}^{*}\right)$$

A Johnson–Cook model with damage parameters was defined for the AA6061-T6 tubing, with damage parameters widely utilized in the open literature [66,67,68], subjected to axial crushing due to the catastrophic fractures observed for this alloy [69,70], discussed further in Section 3. No failure algorithms were required for the cutting models due to the Eulerian element formulation mentioned in Section 2.4.1, or for extruded tubing in a T4 temper condition which did not experience brittle fracture.

#### 2.4.3. 4140 Steel Fixture Material Model

The remaining metallic, hardened 4140 steel fixtures (e.g., platens, cutting tool) were modeled as rigid entities with an assumed elastic modulus, density and Poisson’s ratio of 207 GPa, 7830 kg/m^3^ and 0.3, respectively. The rigid material assumption was necessary to mitigate the computation requirements of each developed model and was assumed to be reasonable since no deformation, tool wear or similar degradation was observed between tests.

#### 2.4.4. Honeycomb Material Model for Out-of-Plane PVC Foam Compression

An orthotropic crushable foam model (*MAT_HONEYCOMB_026 in LS-DYNA^®^, Livermore Software Technology Corporation, Livermore, CA, USA)) was utilized to replicate the quasi-static compressive stress/strain responses of the H-series PVC foam cores with the data originally presented in Section 2.1.2. The compressive stress/strain responses were fitted to the Avalle model to omit the post-yield stress drop-off, consistent with [57].

#### 2.4.5. Semi-Empirical Material Model for in-Plane PVC Foam Compression

A previous investigation by Magliaro et al. [57] concluded that H200 and H250 PVC foam cores experience pronounced volumetric strains, $\varepsilon_{V}=\varepsilon_{RD}+\varepsilon_{IPD}$, due to in-plane expansion during axial collapse, associated with non-zero Poisson’s ratios (recall Table 1), which are constricted by the inner walls of the tubing subjected to cutting. A schematic and brief explanation of the underlying mechanisms is provided in Appendix A. This in-plane foam expansion cannot be modeled utilizing current releases of commercially available finite element software packages, including LS-DYNA^®^, since the standard numerical models available for structural foams demand a zero-value for the Poisson’s ratio. A translational nonlinear spring was introduced between the foam cores subjected to 10-bladed cutting/clamping (originally shown in Figure 4b), with the volumetric strain due to in-plane deformation, *ε_IPD_*, related to the crosshead displacement, *x* [57]:(4)$$\varepsilon_{IPD}={\left(\frac{\nu_{foam}}{L-h_{FC}}x\right)}^{2}$$
where the volumetric strain in the rise direction can be stated as $\varepsilon_{RD}=\frac{x}{L}$. Substituting Equation (4) into the Avalle model from Equation (2) and applying the appropriate material parameters from Table 1 yields an expression for the compressive stress due to in-plane foam restriction. Accounting for the total foam core height, *L*, and cross-sectional area of the foam filler, $A_{c}=\pi r_{i}^{2}$, results in the following force/displacement response due to annular restriction of the foam core expansion:(5)$$F_{IPD}\left(x\right)=\frac{\pi r_{i}^{2}}{2}\left[K_{p,RD}\cdot \left(1-e^{-\frac{xE_{RD}}{\left(L-h_{FC}\right)K_{p,RD}}{\left[\nu_{foam}^{2}\left(\frac{x}{L-h_{FC}}\right)\right]\left[1-\left(\frac{x}{L-h_{FC}}\right)\cdot \left[\nu_{foam}^{2}\left(\frac{x}{L-h_{FC}}\right)\right]\right]}^{m_{RD}}}\right)+K_{h,RD}{\cdot \left(\frac{x\left[\nu_{foam}^{2}\left(\frac{x}{L-h_{FC}}\right)\right]}{L-h_{FC}-x\left[\nu_{foam}^{2}\left(\frac{x}{L-h_{FC}}\right)\right]}\right)}^{n_{RD}}\right]$$

Note that the force was halved due to symmetry conditions. This expression was applied as the nonlinear force response in the translational spring to account for the axial force associated with in-plane foam core restriction, which was omitted in recent modeling efforts [55] and thus represents an advancement in the coupled Lagrangian/Eulerian model.

#### 2.4.6. Boundary Conditions

The compounded energy dissipation systems were simulated utilizing a double precision massive parallel processing (MPP) version of LS-DYNA^®^, release R12.1 (Livermore Software Technology Corporation, 374 Las Positas Road, Livermore, CA, USA). Compression of the composite AA6061/PVC foam structures was simulated by positioning a rigidly fixed platen at the upper ends of the apparatus and applying a 1 mm/ss displacement rate to the lower platen (crushing) and cutter assembly (cutting). Half-symmetry was invoked for the 10-bladed cutting/clamping model shown in Figure 4b, given the symmetric nature of the cutting profile noted in Section 3, and to mitigate the computational requirements of the Eulerian element formulation. Symmetry conditions were not implemented for the axial crushing mode (Figure 4a) due to the erratic nature of lobe formation and brittle fracture. Segment-based contact algorithms were applied between all Lagrangian entities with static and kinetic friction coefficients of 0.35 and 0.30, respectively, at the aluminum/steel interfaces and 0.25 and 0.20, respectively, at the contact interfaces. A fluid/structure interaction approach was applied for the cutting simulations to model contact between the AA6061 tubing and steel fixtures with the stated kinetic friction coefficients.

#### 2.4.7. Validation

The numerically predicted force/displacement responses, $f\left(x\right)$, were validated utilizing complementary experimental test results, $F\left(x\right)$, by calculating the cumulative error, *C*, defined in Equation (6), with an anticipated ideal value of zero [71]. This metric assessed deviations between the theoretical and experimental collapse forces over the entire displacement domain (0, *δ_T_*). Average cumulative errors of 0.057 and 0.190 were calculated for the axial cutting and 10-bladed cutting/foam crushing deformation modes, respectively. These values were consistent with approximately 88% correlation between the numerical modeling and experiments.
(6)$$C=\frac{1}{\delta_{T}}\int_{0}^{\delta_{T}}\left|\frac{F\left(x\right)-f\left(x\right)}{f\left(x\right)}\right|dx$$

### 2.5. Performance Metrics

Several performance metrics were utilized to evaluate the experimental and theoretical results. The total energy absorbed, *TEA*, was defined as
(7)$$TEA=\int_{0}^{\delta_{T}}F\left(x\right)dx$$
where $F\left(x\right)$ and *δ_T_* represent the reaction force during axial collapse at a given displacement, *x*, and the total displacement, respectively. The specific energy absorbed, *SEA*, per unit mass of the sacrificial structure is defined as Equation (8); the mass, *m_EA_*, of a circular tubing packed with solid foam cores can be calculated utilizing Equation (9).
(8)$$SEA=\frac{TEA}{m_{EA}}$$
(9)$$m_{EA}=\pi \left[\rho_{EM}\left(r_{o}^{2}-r_{i}^{2}\right)L+\sum_{j=1}^{k}{\left(\rho_{FC}r_{i}^{2}h_{FC}\right)}_{j}\right]$$
where the subscripts ‘EM’ and ‘FC’ denote the extruded material and foam core, respectively, assuming *k* = 0 if hollow and *k* = 8 if foam-filled. The mean force, *F_m_*, over the force/displacement response was determined as follows:(10)$$F_{m}=\frac{1\;}{\delta_{T}}\int_{0}^{\delta_{T}}F\left(x\right)dx$$

The compressive force efficiency, *CFE*, quantifies the fluctuation between the average and the peak reaction forces:(11)$$CFE=\frac{F_{m}}{F_{max}}$$

The energy-absorbing effectiveness factor (EAEF), *ψ*, quantifies the *TEA* capacity, structural stability and material utilization for a sacrificial structure normalized with respect to the *TEA* for an equal volume of material, ${\bar{V}}_{EA}$, in uniaxial tension [72]:(12)$$\psi =\frac{F_{m}\delta_{T}}{\pi \varepsilon_{f}L\left[\sigma_{o}\left(r_{o}^{2}-r_{i}^{2}\right)+\sigma_{P,foam}\sum_{j=1}^{k}{\left(r_{i}^{2}h_{FC}\right)}_{j}\right]}$$
where *σ_o_* and *σ_P,foam_* are the flow stress of the extruded material and foam plateau stress, respectively, and *ε_f_* is the extruded material fracture strain.

### 2.6. Analytical Modeling

A parametric study was also conducted for the AA6061/PVC foam energy absorbers to assess the influence of key engineering parameters, especially critical tube/foam dimensions, on the efficacy of these composite structures subjected to the novel 10-bladed cutting and traditional axial crushing modes and is presented in Section 4.5, following an assessment of the experimental and numerical results. This study was completed utilizing well-established and validated analytical models. The material-related input data (flow strengths, *σ_o_*, blade friction coefficients, *μ_b_*, etc.) were originally provided in Section 2.1, Section 2.4.2 and Section 2.4.3. Only the most critical aspects of these models (i.e., those necessary to replicate the calculations) are presented.

#### 2.6.1. Mean and Peak Axial Crushing Forces

The plastic deformation mechanisms for foam-filled metallic tubing were considered in numerous theoretical studies that linked axisymmetric lobe formation in unfilled structures [72,73] with foam core compression and increased resistance to hinge bending [58,74]. The mean force, *F_m,crush_*, experienced by foam-filled tubes can be predicted as
(13)Fm,crush=2πσot2ro2tcos−1 3ρfρs+2sin cos−1 3ρfρs1−3ρfρs+1+3πri2σY,s10·ρfρs32
where *ρ_f_* and *ρ_s_* are the density of a given foam configuration and its base material, respectively.

The peak crushing force, $F_{crush,max}$, for the AA6061/PVC foam structures is dominated by lobe formation and out-of-plane foam compression. The PVC foam plateau stress, *σ_P_*, can be related to the density, *ρ_s_*, and yield strength, *σ_Y,s_*, of the base material [63]:(14)σP=1.82·σY,sρfρs32
where density and yield strength of PVC are 1380 kg/m^3^ and 55 MPa, respectively. The peak crushing force associated with initial lobe formation is the product between the flow strength and cross-sectional area of the tubing, $F_{peak}=\pi \left(r_{o}^{2}-r_{i}^{2}\right)\sigma_{o}$, estimated as
(15)Fcrush,max=πro2−ri2σo+1.82·ri2σY,sρfρs32

These models were previously validated for AA6061 tubing in both T6 and T4 temper conditions [58]; brittle fracture in the former was concluded to represent a negligible contribution toward *TEA* since the lobes form and collapse with complete/intact hinges before fracturing.

#### 2.6.2. Compounded Cutting/Foam Crushing Forces

The accepted standard approach for analytically modeling unfilled [46,75] and PVC foam-filled [57,58] AA6061 tubes subjected to axial cutting and cutting/clamping involves the prediction of key points from the force response to predict the entire force/displacement response. Readers are encouraged to seek out the mentioned references for thorough, step-by-step derivations and discussions of the underlying theory. The total reaction force associated with a compounded cutting/foam crushing deformation mode, *F_cfc_*, can be generically stated as [57]
(16)$$F_{cfc}=F_{cut}+F_{foam}+F_{inter}=n_{b}\left[1+\mu_{b}\cot\;\theta \cos^{2}\;\left(\frac{\sqrt{{\Delta r}_{ss}\left({2R}_{a}-{\Delta r}_{ss}\right)}}{\left(R_{a}-{\Delta r}_{ss}\right)}\right)\right]\cdot F_{P}+F_{foam}+F_{inter}$$
where an *n_b_*-bladed cutter imparts resistive forces due to plastic deformation, *F_P_*, which also influence the frictional forces at the cutting interface, and the foam cores experience axial compression, *F_foam_*, and interactions with the inner wall of the tubing, *F_inter_*. The remaining geometric parameters are introduced in Appendix A and Appendix B. The forces associated with tube cutting, *F_cut_*, present with elastically dominated wedge indentation, *F_E_*, followed by plastic wedge penetration, *F_w_*, and steady-state cutting/clamping, *F_ss_*:(17)$$F_{cut}\left(x\right)=\left\{\begin{array}{c} F_{E}\left(x\right),0\leq x<\delta_{Y}\;\\ F_{w}\left(x\right),\delta_{Y}\leq x<l_{b}\;\\ F_{ss}\left(x\right),x>l_{b} \end{array}\right.$$
where the indentation, wedge penetration and steady-state forces can be predicted with respect to the axial displacement, *x*, utilizing Equation (18) through Equation (20), respectively [46].
(18)$$F_{E}\left(x\right)=\frac{n_{b}E_{EM}}{L}\left(2R_{r}\cos\;\theta +t\right)\left(2x\;\tan\;\theta +T\right)x$$
(19)$$F_{w}\left(x\right)=\frac{F_{E}\left(\delta_{Y}\right)-F_{W}}{{\left(\delta_{Y}-l_{b}\right)}^{2}}\left[x^{2}-2l_{b}x+l_{b}^{2}\right]+F_{W}$$
(20)$$F_{ss}\left(x\right)=n_{b}\left[1+\mu_{B}\;\cot\;\theta \;\cos^{2}\;\left(\frac{\sqrt{{\Delta r}_{t}\left({2R}_{a}-{\Delta r}_{t}\right)}}{\left(R_{a}-{\Delta r}_{t}\right)}\right)\right]\cdot \left(F_{p}+\frac{{E_{EM}I}_{G_{P}}{∆}_{v}\left(h_{f}-{\mu_{B}l}_{f}\right)}{\left(1-\nu^{2}\right)X_{P}}\right)$$

The forces associated with PVC foam compression and tubing/foam interactions include the bulk out-of-plane resistance to crushing by the closed foam cells, in-plane/volumetric resistance from the inner walls of the tubing and relative sliding at the interface [57], expressed together as
(21)$$\begin{array}{c} F_{foam}\left(x\right)\\ +F_{inter}\left(x\right) \end{array}=\begin{array}{c} \begin{array}{c} \pi r_{i}^{2}K_{p,RD}\cdot \left[1-e^{-\frac{xE_{RD}}{\left(L-h_{FC}\right)K_{p,RD}}\left(\frac{L-h_{FC}+\nu_{foam}^{2}x}{L-h_{FC}}\right){\left(\left(\frac{L-h_{FC}-x}{L-h_{FC}}\right)\cdot \left(\frac{L-h_{FC}+\nu_{foam}^{2}x}{L-h_{FC}}\right)\right)}^{m_{RD}}}\right]\\ +\pi r_{i}^{2}K_{h,RD}{\cdot \left[\frac{x\left(\frac{L-h_{FC}+\nu_{foam}^{2}x}{L-h_{FC}}\right)}{L-h_{FC}-x\left(\frac{L-h_{FC}+\nu_{foam}^{2}x}{L-h_{FC}}\right)}\right]}^{n_{RD}} \end{array}\\ +2\pi \mu_{f}r_{i}\left(L-h_{FP}-x\right)\left[K_{p,IPD}\cdot \left(1-e^{-\frac{\nu_{foam}E_{IPD}x}{{LK}_{p,IPD}}{\left(\frac{L+\nu_{foam}x}{L}\right)}^{m_{IPD}}}\right)+K_{h,IPD}{\cdot \left(\frac{\nu_{foam}x}{L-\nu_{foam}x}\right)}^{n_{IPD}}\right] \end{array}$$

The mechanical properties of the various PVC foam cores in the rise and in-plane directions (RD and IPD, respectively) were initially given in Section 2.1.2, and any remaining geometric parameters are provided in Appendix B. The maximum reaction force can be determined utilizing Equation (16) by assuming this value coincides with the maximum compression distance, *δ_max_*, such that $F_{cut,max}=F_{cfc}\left(\delta_{max}\right)$.

#### 2.6.3. Scope of Investigation

The parametric scope for this theoretical study, summarized in Table 4, was selected for tube diameter/wall thickness pairs that conform to the minimum and maximum allowable sizes for the proposed apparatus previously shown in Figure 3. The composite AA6061/PVC foam structures were assumed to possess a 200 mm length and experience 125 mm of quasi-static compression (approximately 63% relative to their height) in the theoretical calculations. Each material pairing/geometric configuration was investigated for the 10-bladed cutting and axial crushing deformation modes utilizing the key performance metrics (namely, *F_m_*, *CFE* and EAEF) defined in Section 2.5 to comprehensively assess the overall efficacy of the novel compounded cutting/foam crushing system.

## 3. Results

The following section summarizes the experimental observations, including the observed deformation profiles, force/displacement responses and average performance metrics (*TEA*, *SEA*, etc.) as defined in Section 2.5. Energy-absorbing capabilities and structural stability during collapse were analyzed to characterize the overall effectiveness of each tube/foam core pair. The complementary numerical simulation results were also presented to directly assess the predictive capabilities of the finite element models.

### 3.1. Deformation Profiles

The tubes in a T6 temper condition were considered first since this represents the higher-strength, as-received case for the 6061 alloy. Representative samples as observed post-test are shown for the unfilled case in Figure 5a,b through Figure 5d for the composite H130, H200 and H250 foam-filled samples, respectively. Repeated brittle lobe fracture was observed in every test case and simulation involving the axial crushing mode; the mechanism was least severe for the unfilled case and worsened (i.e., cracking was more rampant and deviated further away from the deformed lobes) as the foam core density increased. The increasingly dense foam cores prevented inward buckling of the deformed lobes, especially during the final 30 mm of displacement, intensifying the brittle fracture.

The complementary tubes subjected to 10-bladed cutting/clamping exhibited reciprocal behavior in terms of their structural stability. The unfilled tubing (Figure 5a) experienced severe material entanglement and lateral shifting as the cutting process reached a steady state and the petalled material exited the cutter. The newly introduced platen and foam cores provided annular support which eliminated the eccentricity that developed for the unfilled tubes. The sustained concentricity between the composite AA6061/PVC structures and the cutting tool further ensured that the petalled sidewalls formed symmetrically. There were no discernible differences between the observed cutting deformation profiles for each foam core density (Figure 5b through Figure 5d), which suggests that the annular support is more significant than the density of the foam cores.

The annealed, T4-tempered tubes exhibited deformation mechanisms associated with a ductile progressive folding deformation mode, especially for the unfilled case shown in Figure 6a which experienced axisymmetric lobe formation. The complementary tubing subjected to 10-bladed cutting/clamping experienced unstable lateral distortion as the cutting process reached a steady state; the extruded material above the blade tips succumbed to excessive hoop stresses caused by twisting/flaring of the petals and the localized cutting phenomena. The stability for the AA6061-T4 extruded tubing with H130 and H200 foam cores, shown in Figure 6b and Figure 6c, respectively, improved during axial collapse for the 10-bladed cutting mode since PVC foam cores introduced an annular constraint which improved concentricity and resisted lateral bowing. In contrast, the H130 and H200 foam-filled tubing subjected to crushing experienced moderately erratic collapse which was characterized by lateral kinking of the tube axis.

The annealed tubing wrapped around the PVC foam cores as they densified, rather than experiencing the brittle lobe fracture observed for the T6 temper condition, promoting this unstable mechanism. The H250 foam-filled tubes, shown post-test in Figure 6d, were less susceptible to lateral kinking due to the in-plane expansion (Poisson effects discussed in Section 2.4.2) of the denser H250 foam which forced the lobe formation to occur outward rather than wrapping around the densified/compacted core. The tubes subjected to cutting exhibited a deformation profile that was visually similar to the other foam-filled material groups, with stable petal formation, concentric flaring and material remaining intact above the blade tips. Overall, the composite AA6061/PVC foam-filled tubing experienced greatly improved stability when subjected to 10-bladed cutting for both T6 and T4 temper conditions while the axial crushing modes exhibited unstable collapse mechanisms which are unsuitable for real-world safety systems.

### 3.2. Force/Displacement Responses

Experimentally and numerically obtained crushing- and cutting-based force responses for the unfilled AA6061-T6 tubes are provided in Figure 7a; the former deformation mode was characterized by repeated fluctuations with large differences between local minima and maxima due to brittle lobe fracture and subsequent reformation. The cutting deformation mode was characterized by a more ideal, near-constant reaction force in the steady-state regime (i.e., >20 mm of displacement, once the deformation profile fully developed) which was maintained for the remainder of each test. Similar behavior was also observed for the H130 foam-filled structures, as shown in Figure 7b, with a 12% increase in the stead-state/mean forces due to the compressive resistance of the foam cores. The force responses for the H200 and H250 foam-filled AA6061-T6 tubing, provided in Figure 7c and Figure 7d, respectively, exhibited progressive increases in the steady-state/mean (overall) reaction forces with increasing displacement due to more severe densification of the foam cores and intensifying tube/foam interactions. Further details on the latter phenomenon are provided in Section 4.3.

The transition from a steady-state cutting force to a nonlinear response, which resembled the densification trends visible in Figure 2, supported the notion that the tubing/foam interactions observed for the compounded cutting/foam crushing system were caused by suppressed in-plane expansion of the densified foam cores by the extruded material which remained intact above the cutter. The axial crushing deformation modes progressed with increasingly severe fractures caused by radial material displacement after each lobe formed. This trend was observed in the force/displacement response as intensifying variations between the local minima and maxima, approximately 27 kN for the unfilled case compared to 34 kN, 39 kN and 43 kN for the H130 through H250 foam-filled cases, respectively, as each lobe separated from the composite structure.

The AA6061-T4 tubing exhibited more consistent behavior between the unfilled (Figure 8a) and PVC foam-filled (Figure 8b through Figure 8d) when subjected to axial crushing. The secondary load fluctuations associated with lobe formation were more comparable between material configurations since the (annealed) extruded material was not prone to the brittle hinge fractures associated with the T6 temper condition. The major distinctions between test cases of increasingly dense foam cores were proportionate scaling of the secondary load valleys and a transition from repeated, monotonic loading to a gradually increasing mean force due to increased axial resistance of the foam to compression and pronounced densification and tube/foam interactions, respectively.

The cutting/clamping (C10) mode was characterized by significantly improved consistency in the force/displacement response, even for the unfilled case presented in Figure 8a, which displayed unstable circumferential profile distortion in Section 3.1. The steady-state compressive force experienced moderate fluctuations, between approximately 5 kN of the mean value, due to this effect. These deviations were nearly eliminated by the presence of the H130, H200 and H250 foam in Figure 8b through Figure 8d, respectively, due to the in-plane annular support provided by these entities. A transition from steady-state to progressive linearly increasing reaction forces consistent with the AA6061-T6 test groups was also observed. The consistency between temper conditions for compounded 10-bladed cutting/foam crushing highlights the versatility of this novel deformation mode compared to the traditional axial crushing mode, which exhibited disparate performance, with separate nonideal energy-absorbing mechanisms that are not desirable for occupant safety systems, between material pairs.

### 3.3. Energy-Absorbing Capabilities

The performance metrics obtained from the experimental force/displacement responses are summarized in Table 5 and Table 6 for the test groups considering AA6061 tubes in T6 and T4 temper conditions, respectively. These data were utilized for the analyses and corresponding discussions provided in Section 4. These metrics reflect the observations from Section 3.1 and Section 3.2; the overall effectiveness (EAEF) of the composite AA6061/PVC foam structures subjected to 10-bladed cutting exceeded the complementary values for the axial crushing mode by 15% to 52%. The enhancement was associated with 5% to 25% increases in *TEA* capacity, 10% to 104% increases in *CFE* and improved structural stability during collapse when comparing similar tubing/foam core pairs.

## 4. Discussion

The following section contains analyses of the findings from Section 3 to characterize and quantify the influence of extrusion/foam core pairing on the energy dissipation characteristics of the proposed composite system. Comparisons between the performance of identical structures subjected to the traditional axial crushing mode and the novel, compounded cutting/foam crushing mode were emphasized. These analyses confirmed generally unfavorable mechanical performance for the composite AA6061/PVC foam structures subjected to crushing with significantly improved stability and insensitivity to the tube/foam pairing for compounded cutting/foam crushing.

### 4.1. Axial Crushing Versus 10-Bladed Cutting/Clamping

The analyses in this section begin with a direct comparison (i.e., considering identical tube/foam material pairs) of the performance metrics obtained for the traditional axial crushing mode to the novel, compound 10-bladed cutting/foam crushing deformation mode. The relative change, *R_DM_*, in a given performance metric, *Q*, for the 10-bladed cutting (C10) mode compared to axial crushing (AC) was calculated for a specific tubing/foam pair as
(22)$$R_{DM}=\left(\frac{Q_{C10}-Q_{AC}}{Q_{AC}}\right)\cdot 100\%$$

The average values for this metric, which were referred to as enhancements since the 10-bladed cutting/clamping mode eclipsed the performance observed for axial crushing, are summarized in Figure 9a,b for the test groups with AA6061 extruded tubing in T6 and T4 temper conditions, respectively. The *TEA* capacity refers to the average enhancement in *TEA* and *F_m_* since the *R_DM_* for these metrics, per the definitions from Section 2.3.2, were near-identical (i.e., the *F_m_* is essentially a scale factor of the *TEA* and thus should experience identical relative changes).

The most dramatically improved performance metric for both temper conditions was the *CFE*, with peak values of 104% and 61% for the unfilled AA6061-T6 and AA6061-T4 tubes, respectively. The enhancements reduced but still remained above 10% for the considered scope, with the inclusion of H200 and H250 PVC foam cores since these configurations promoted tubing/foam interaction effects which disproportionately influenced the peak cutting forces. The overall effectiveness (EAEF), *ψ*, experienced an opposing trend of greater enhancements for the foam-filled configurations compared to the hollow tubes, with average values of 55% and 26%, respectively. The improvements in effectiveness were greater for the AA6061-T6 test groups, reflecting the ability of the novel 10-bladed cutting to eliminate the extensive brittle fractures observed for axial crushing. The increased *TEA* capacity and improved lateral stability for test groups considering AA6061-T4 tubing yielded 10% to 25% lower enhancements in effectiveness since the lateral kinking observed during the axial crushing test was less detrimental than the brittle fracture associated with the (higher-strength) T6 temper condition.

### 4.2. Constitutive Material Pairing

#### 4.2.1. Influence of Extruded Tube Temper Condition

The temper condition of the extruded tubing significantly influenced the stability of the composite AA6061/PVC foam energy absorbers subjected to crushing, with structures containing the T4 temper condition influenced by uneven foam compression and subsequent lateral kinking. Sectioned views of a representative specimen post-test and the corresponding simulated profile are provided in Figure 10a; for brevity (considering the general consistencies observed in Section 3.1), only the T4/H130 group is shown. The principal deformation mechanisms are visibly apparent and were accurately replicated by the finite element models, including in-plane compression of the foam, uneven axial compression and lateral distortion of the tube axes. The complementary case containing AA6061-T6 tubing is shown in Figure 10b; note that sectioned views could not be obtained due to the fragility of the post-test specimen. Unloading and subsequent elastic springback of the PVC foam cores revealed the extent of the brittle fracture and material separation observed for energy absorbers in these groups. While the annealed samples are more stable, both configurations of the composite AA6061/PVC foam structures exhibit highly unstable collapse mechanisms which are ill-suited to occupant safety systems.

The mentioned improvements in structural stability for the 10-bladed cutting/foam crushing deformation mode and consistency between T4 and T6 temper conditions are evident from an examination of the deformed cross-sections provided in Figure 11a,b. The post-test deformation profiles are nearly indistinguishable between temper conditions and can only be discerned by the reduced luster of the AA6061-T4 extruded tubing due to the heat treatment. Both composite structures quickly achieved a steady-state deformation profile which was maintained for the entire duration of each deformation event, exemplified by the symmetric nature of the petalled sidewall material and consistent foam core compression. The high degree of consistency between the composite structures with the as-received and annealed AA6061 tubing, especially the ability to eliminate uncontrolled fracture in the former, is a unique capability of the proposed compounded cutting/foam crushing system which is rarely achieved for traditional crushing-based safety systems with lightweight materials [1,2,6,21,36].

The relative change, *R_EM_*, in a given performance metric, *Q*, for energy absorbers composed of AA6061 tubes in the annealed, T4 temper condition compared to the as-received T6 temper condition can be quantified as
(23)$$R_{EM}=\left(\frac{Q_{T6}-Q_{T4}}{Q_{T6}}\right)\cdot 100\%$$

The average values are summarized in Figure 12a,b for AA6061 energy absorbers, with and without PVC foam cores, subjected to axial crushing and 10-bladed cutting/clamping, respectively. The annealed tubing exhibited dramatically improved *CFE*s, between 10% and 27%, due to the elimination of brittle lobe fracture. The *TEA* capacity of test groups composed of AA6061-T4 tubing was reduced, however, by only 9% on average, in contrast to the 35% difference between flow strengths for the T6 and T4 temper conditions. The tubes subjected to 10-bladed cutting/clamping exhibited more consistent behavior, with little difference (i.e., <5% deviation) between *CFE* values when comparing temper conditions, further confirming the insensitivity of this deformation mode to the material pair. The EAEF and *TEA* capacity were reduced by 12% to 21% for the lower-strength T4 temper condition, with little variation between test groups, since brittle fracture did not occur for AA6061-T6 tubing subjected to the novel cutting/foam crushing deformation mode.

#### 4.2.2. Influence of PVC Foam Core Density

The relative contributions associated with the presence of the PVC foam cores, *R_FC_*, were considered for a given test group/material pairing by assessing the evolution in each performance metric for the foam-filled tubing with respect to the unfilled, reference test case:(24)$$R_{FC}=\left(\frac{Q_{FC}-Q_{Unfilled}}{Q_{Unfilled}}\right)\cdot 100\%$$

The evolutions in this metric with increasing foam core density are summarized in Figure 13a,b for AA6061 extruded tubing in T6 and T4 temper conditions, respectively, subjected to axial crushing. The complementary trends are plotted in Figure 13c,d for the 10-bladed cutting/clamping deformation mode. The *TEA* capacity increased by approximately 81% for the H250 foam-filled cases compared to the unfilled tubing, for the 10-bladed cutting mode compared to a corresponding 96% relative increase for the axial crushing mode. This relative difference was expected since the 10-bladed cutting/clamping mode was associated with higher energy-absorbing capacity (i.e., the cutting of AA6061 tubes contributed toward a greater portion of the *TEA*), discussed further in Section 4.4.

The *CFE* experienced a moderate decrease of approximately 20% for H250 foam-filled tubing subjected to the 10-bladed cutting/clamping (Figure 13c,d) due to the onset and intensification of tube/foam interactions which led to earlier densification of the foam cores with elevated forces in the final 20 mm of displacement. Furthermore, the *CFE*s for the unfilled tubing subjected to cutting regularly exceeded 90% (Section 3.2), which left little margin for relative enhancements. This metric experienced consistent 20% to 30% improvements for the complementary tubes subjected to axial crushing (Figure 13a,b) as the fluctuations associated with lobe formation were offset by the steady-state foam compression. The energy-absorbing effectiveness was reduced for the H130 foam-filled tubes subjected to axial crushing, due to the instabilities mentioned in Section 3.1 and Section 3.2, and recovered to near-equivalent effectiveness for the H200 foam-filled cases and marginally outperformed the hollow tubing by 5% to 15% for the high-density H250 configurations.

The composite structures subjected to 10-bladed cutting/clamping experienced an enhancement in EAEF, *ψ*, of 28% for the H250 foam-filled configurations, compared to the reference (hollow) cases, since the collapse mechanisms remained stable for the considered tube/foam material pairs. Relative enhancements in the EAEF with increasing foam core density were highly similar between structures composed of T6 and T4 tempered extrusions (Figure 13c,d, respectively) since the observed deformation mechanisms were visually near-identical, leading to similar utilization of the sacrificial material volume. The corresponding relative changes in *SEA* were negligible with a quasi-linear relative increase of 10% for the H250-filled tubing with respect to the unfilled cases since the tubing contributed toward a greater extent of the *TEA* capacity. In contrast, the relative increases in EAEF and *SEA* were near-identical for the AA6061 tubing in T6 and T4 temper conditions (Figure 13a,b, respectively) subjected to axial crushing. The similarity between these trends was attributed to the positive influence of the PVC foam cores on the mass-specific energy absorption, since these entities contributed more significantly toward the TEA capacity than the AA6061 tubing, and the competing negative effects caused by worsening brittle fracture and lateral kinking of the extrusion axes (Section 3.1 and Section 3.2).

### 4.3. Extruded Tubing/Foam Core Interactions

Comparisons between the mechanical responses for hollow (unfilled) and H130, H200 and H250 foam-filled AA6061-T6 (as-received) tubing are provided in Figure 14a through Figure 14c, respectively, for the 10-bladed cutting/clamping deformation mode, similar comparisons for the axial crushing mode are shown in Figure 14d through Figure 14f. The former deformation mode was characterized by negligible tubing/foam core interactions for the H130-filled case and moderate linear increases which deviated from the steady-state algebraic sum for the H200- and H250-filled configurations. The extruded tubing subjected to crushing presented erratic interaction effects since in-plane foam compression occurred during hinge collapse, which was released when brittle fracture initiated, accounting for the sporadic onset of tubing/foam interactions. This effect was most pronounced for the H200 and H250 foam-filled configurations, especially above 75 mm of displacement, coinciding with foam densification and subsequently pronounced interference contact forces between the deformed lobes and foam.

The interaction effects for composite structures composed of AA6061-T4 tubing subjected to 10-bladed cutting/clamping are provided in Figure 15a through Figure 15c. The general trend was consistent with the observations for the samples with T6 temper conditions; interactions were negligible for the H130-filled group since this material exhibits a near-zero Poisson’s ratio and thus only experiences axial compression above the blade tips, accomplishing the desired combination of steady-state reaction forces. The composite structures with H200 and H250 foam cores also displayed progressive deviations from the steady-state values. The complementary tubes subjected to crushing, presented in Figure 15d through Figure 15f, exhibited large fluctuations with secondary peak forces that exceeded the initial value due to tube/foam interference during collapse. The mechanical performance was improved compared to the crushed specimens composed of AA6061-T6 tubing since brittle fracture was eliminated for the T4 temper condition and thus allowed for continuous lobe formation. However, the interaction effects were still highly erratic and presented with uncertainties and non-desirable characteristics, including the tendency for the force to reduce below the value anticipated with the algebraic sum.

### 4.4. Material Contributions toward Total Energy Absorption

The total energy absorption for the composite AA6061/PVC foam structures in the present study can be generally expressed in terms of the following principal deformation mechanisms, *E_i_*:(25)$$TEA=\sum E_{i}=E_{C10/AC}+E_{foam}+E_{inter}$$
where the terms represent, from left to right, either 10-bladed cutting/clamping or axial crushing of the extruded 6061 tubing, compression of the PVC foam cores and the related interaction effects, respectively. The individual energy absorption terms can be estimated utilizing the data initially provided in Section 3.3 and Section 4.3, with the foam-related terms omitted for the hollow cases. These values can be normalized to assess the relative contribution of each energy dissipation mechanism, *η_E_*, defined in Equation (26), such that $\sum {\eta_{E}}_{i}=\eta_{C10/AC}+\eta_{foam}+\eta_{inter}=1$.
(26)$$\eta_{E}=\frac{E_{i}}{TEA}\cdot 100\%$$

The relative contributions of each principal deformation mechanism for the hollow and foam-filled AA6061 extruded tubing subjected to axial crushing (AC) and 10-bladed cutting/clamping (C10) are summarized in Figure 16 and Figure 17, respectively.

The axial compression of the foam cores presented with quasi-linear increases in the relative contribution toward TEA capacity which was nominally proportionate to the increasing density. For example, comparing the H130 and H250 foam-filled configurations, the contribution due to foam compression, *η_foam_*, experienced a 78% relative increase across all test configurations with respect to an 86% increase in density. Relative contributions due to tube/foam interactions were notably less consistent between test groups. The composite structures composed of AA6061-T6 tubing presented with an initially marginal average contribution of 8% when subjected to axial crushing for the H130 foam-filled case and increased to 14% between the H200 and H250 groups, as shown in Figure 16a. The reduced influence of interactions for the former case was associated with Poisson’s effects for the lower-density configuration, allowing for more inward hinge formation and severe brittle lobe fracture with in-plane damage to the foam cores, originally shown in Figure 10b.

The annealed, T4 temper condition (summarized in Figure 16a) did not experience brittle fracture and thus tubing/foam interactions were highly consistent between each foam-filled test case, with an average relative contribution of 1% toward TEA capacity when subjected to crushing. The contributions were more consistent for the 10-bladed cutting mode for composite structures composed of both T6 and T4 temper conditions, shown in Figure 17a and Figure 17b, respectively. The average TEA capacity associated with interaction effects was approximately 30% higher for a given AA6061-T4/PVC foam-filled group compared to a complementary structure composed of the T6 temper condition due to the higher strength of the extruded material of the former. However, the consistency between relative increases was expected since both configurations remained intact above the blade tips in the region containing the foam cores (recall Section 3.1 and Section 4.2.1), and thus identical interaction mechanisms were observed despite the disparities between extruded material configurations.

### 4.5. Theoretical Study

The following results were obtained utilizing the established, accepted analytical models originally presented in Section 2.6. The ratio of relative change, *R_Q_*, in a given performance metric, *Q*, was calculated for each of the previously listed AA6061 extruded tubing/PVC foam pairings to quantify the performance of the proposed, novel energy dissipation system to the traditional, reference case:(27)$$R_{Q}=\left(\frac{Q_{C10}}{Q_{AC}}-1\right)\cdot 100\%$$

Note that the comparison ratios for the *TEA* and *SEA* would be identical since the comparisons between cutting and crushing considered geometrically identical composite structures for each case, and thus this ratio was only calculated and presented for the mean forces, *F_m_*. The relative change ratios quantifying the change in *TEA* capacity for 10-bladed cutting/clamping versus crushing are provided in Figure 18a through Figure 18c and Figure 18d through Figure 18f for H130, H200 and H250 foam-filled AA6061 tubes in T6 and T4 temper conditions, respectively.

The mean force anticipated for the 10-bladed cutting/clamping mode was either near-equivalent (i.e., within 5% of the *TEA* capacity observed for crushing) or exceeded by significant margins for the H130 foam-filled configurations. The relative enhancements in *TEA* capacity for composite AA6061/PVC foam structures subjected to 10-bladed cutting versus axial crushing increased, in all theoretical groups, with decreasing wall thickness and tube diameter. The mean force increased by a factor of 2 or greater when considering the lowest diameter/thickness pairs in all groups (44.5 mm OD, 0.79 mm wall) and was marginally lower, between 5% and 20% reductions, for the groups with the largest tube geometry (63.5 mm OD, 3.18 mm wall). High sensitivity to thin wall thicknesses and larger diameters, especially the trend of relative comparison metrics approaching equivalence with increasing tube profile/cross-section size, suggests that a greater presence of internal PVC foam cores diminishes the enhancing effect on *TEA* capacity. Furthermore, the relative enhancements were lower for tubes in the T6 temper condition since this material configuration experiences mean axial crushing forces which are not hindered/reduced by strain hardening, in contrast to the annealed T4 temper condition.

The *CFE* experienced more dramatic enhancements, with peak values of the *CFE* ratio reflecting improvements of this metric by factors between 2 and 3, with the full scope plotted in Figure 19. Such dramatic enhancements were expected since the 10-bladed cutting/clamping mode was experimentally shown in Section 3.2 and Section 4.3 to eliminate the large, unfavorable load fluctuations associated with axial crushing. The enhancements were greater for the composite structures composed of AA6061-T6 tubing, shown in Figure 19a through Figure 19c for the H130, H200 and H250 foam-filled cases, respectively, compared to the complementary AA6061-T4 structures (Figure 19d through Figure 19f) since the former structures experienced brittle fracture when subjected to axial crushing. However, increasing the tube diameter (i.e., maximizing the foam core size and tube/foam contact area) for both extrusion configurations promoted more significant interactions for the H200 and H250 foam-filled configurations.

This phenomenon, combined with the tendency for the 10-bladed cutting clamping force to transition from a steady state to monotonically increasing with increased foam core densities, led to reductions from the near-ideal *CFE*s (>90%) anticipated for the 10-bladed cutting/clamping mode to absolute values well below 80%. The highest-density, H250 foam-filled composite structures correspondingly exhibited *CFE* values that were near-identical between the cutting and crushing deformation modes for the T6 temper condition (Figure 19c) and reduced by approximately 10% for the T4 temper condition (Figure 19f) for tube diameters and thicknesses exceeding 50.8 mm and 2.79 mm, respectively. Combined with the observations on *TEA* capacity in Figure 18, these configurations represent limited cases where the mechanical performance is comparable between these deformation modes. However, most configurations of the composite AA6061/PVC foam structures experienced enhancement in *TEA* capacity and/or *CFE* when subjected to 10-bladed cutting, especially when considering lower-density foams.

The overall effectiveness (EAEF) ratio of the composite structures subjected to 10-bladed cutting/clamping was found to increase compared to the traditional axial crushing mode for every considered case of AA6061-T6 tubing with H130, H200 and H250 foam fillers, illustrated by Figure 20a through Figure 20c, respectively. Similar trends were observed for the complementary structures composed of AA6061-T4 tubes in Figure 20d through Figure 20f, respectively. Consistent with the previously considered metrics, the relative enhancements were greatest for the lowest diameter/wall thickness pairs. Structures with the largest tube diameter (63.5 mm sample groups) and paired with the H250 foam experienced marginal enhancements in EAEF, generally between 10% and 25%, since the higher-density (and larger) foam cores contributed more significantly toward the TEA capacity and thus diminished the influence of the extruded AA6061 tubing. The universal enhancement primarily occurred due to the increased structural stability of the 10-bladed cutting/clamping mode during collapse, which enabled greater load-bearing capacities, maximum compressed lengths and improved CFEs, quantified together by the noted improvements in EAEF.

### 4.6. Comparisons to Alternative Technologies

The data from the current study, obtained (in part) with a thoroughly validated analytical model [57], were supplemented with experimental evidence utilized to estimate a range of SEA values corresponding to the mean forces to compare with alternative foam-filled energy absorbers. These comparisons considered other lightweight, foam-filled energy absorbers and are visualized with an Ashby plot in Figure 21. The proposed 10-bladed cutting/foam crushing system from the present study offers superior mechanical performance to many previously investigated composite structures composed of aluminum alloys and fiber-reinforced polymers.

Recall from the previous subsection that the composite AA6061/PVC foam structures subjected to the compounded cutting/clamping deformation mode can also eclipse the mechanical performance of complementary structures subjected to axial crushing, which represents the current state of the art. A broad range of mean forces and specific energy absorptions were noted, with upper limits exceeding 100 kN and 50 kJ/kg, respectively. These high-capacity metrics can be achieved with ideal, near-constant reaction forces, and furthermore, the desired mechanical response, including the distribution of energy dissipation between the (cut) tubing and (crushed) foam cores, can be precisely engineered utilizing the numerical and theoretical modeling approaches from Section 2.4 and Section 2.6, respectively.

## 5. Conclusions

The impetus of this study was to assess the influence of extruded/foam core material pairing on the energy-absorbing capabilities, structural stability and overall effectiveness of a novel compounded axial cutting/foam crushing energy dissipation system utilizing composite AA6061/PVC foam structures, with complementary axial crushing experiments to enable direct comparisons to the current state of the art. The tubing was considered in as-received and annealed configurations to observe the potential extent and severity of brittle fracture during collapse, especially when interacting with newly considered, high-density (i.e., 0.10 > relative density) foam cores. The experimental testing and numerical modeling identified trends of increased energy-absorbing capacity and worsening structural stability with increasing foam core density for the AA6061/PVC foam structures subjected to axial crushing. The tubes with T6 temper conditions experienced brittle hinge fracture during collapse with a correspondingly more erratic force/displacement response. The observed fractures were generally more severe when PVC foam cores were added since lobe formation could not occur within the inner profile, diminishing the benefits that would be anticipated for this higher-strength material configuration. The complementary tubing in an annealed, T4 temper condition mitigated the lobe fracture but experienced lateral kinking and unstable shifting of the tube axes.

In contrast, the experimentally observed and numerically predicted deformation mechanisms for the 10-bladed cutting mode were highly consistent, exhibited no evidence of unstable/brittle fracture which is often characteristic of sacrificial aluminum structures and were nearly indistinguishable between each test group. This insensitivity to the extruded material/foam core pairing is a unique capability of the compounded cutting/foam crushing system, which is highly advantageous to structural safety applications, especially since the notoriously brittle nature of lightweight alloys can be effectively eliminated. The performance metrics observed for the 10-bladed cutting/clamping mode eclipsed corresponding values obtained for the axial crushing mode, with average relative enhancements in the *TEA* capacity, *CFE* and overall effectiveness (EAEF) of 14%, 44% and 52%, respectively, over the considered scope. The compounded cutting/foam crushing mode also efficiently mitigated erratic interaction effects since the extruded material remained intact above the cutters during collapse, promoting more consistent interactions.

A theoretical parametric study considering 294 individual material/geometric configurations promoted comprehensive comparisons between the compounded cutting/clamping and axial crushing deformation modes. The composite structures subjected to 10-bladed cutting/clamping could promote up to 90% and 185% enhancements in the mean force, *F_m_*, and compressive force efficiency, *CFE*, respectively, compared to the axial crushing mode. The improvements in *TEA* capacity were greater for composite structures composed of AA6061-T4 tubing since the annular support provided by the PVC foam cores promoted a higher-strength cutting response from this annealed alloy while the complementary T6 temper condition experienced greater *CFE* enhancements due to the elimination of brittle fracture. With a limited scope of geometries, particularly H250 foam-filled tubing with outer diameters exceeding 50.8 mm, the efficacy of the proposed 10-bladed cutting/foam crushing system was diminished. These limitations were associated with pronounced tube/foam interactions, which eliminated *CFE* enhancements, and excessive energy dissipation by the foam cores, reducing the role of the extruded material on *TEA* capacity. An Ashby plot summarizing the present results with respect to alternative technologies demonstrated the potential for the proposed compounded cutting/foam cutting system to significantly extend the loading envelope (i.e., improve mean force and *SEA* capacities) for lightweight safety systems.

To conclude, this study highlighted the versatility of the proposed 10-bladed cutting/foam crushing energy dissipation system by considering multiple geometries, extruded tubing/foam core material pairs and a thorough comparison to the traditional axial crushing deformation mode. The analyses are supported by experimental, numerical and analytical evidence which, most significantly, demonstrate that the compounded cutting/foam crushing system can achieve high-capacity energy absorption with idealized, near-constant reaction forces for a broad range of sacrificial material selections with disparate constitutive properties. This latter point is bolstered by the unique ability of this apparatus to absorb energy without imparting widescale brittle fracture on the lightweight, AA6061 extruded materials despite their moderate ductility. This novel ability to promote more stable collapse in lightweight materials is especially critical since many proposed energy dissipation modes cannot reliably transition away from traditional (e.g., steel alloy) materials due to this previous limitation. Therefore, these findings represent an important contribution toward the current state of the art which can more effectively enable the adoption of lightweight constitutive materials within sacrificial safety systems across numerous industrial sectors.

## Figures and Tables

**Figure 1 materials-16-06282-f001:**
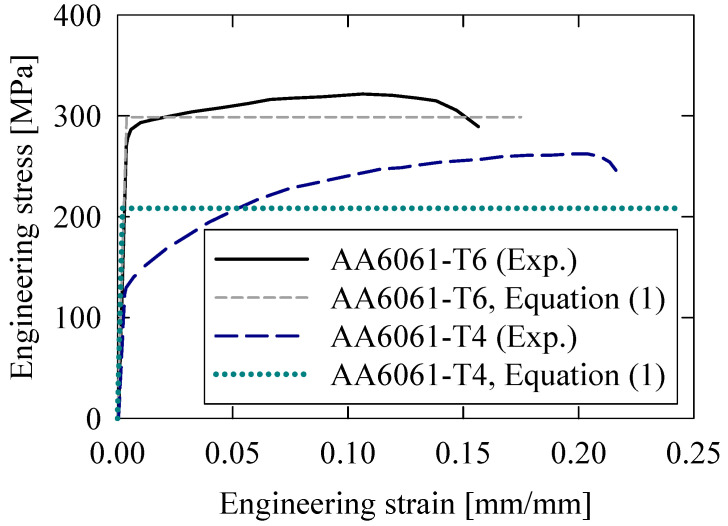
Engineering stress/strain response for extruded AA6061 tubing, in T6 (as-received) and T4 (annealed) temper conditions, with their corresponding equivalent (average) flow stresses (Equation (1)).

**Figure 2 materials-16-06282-f002:**
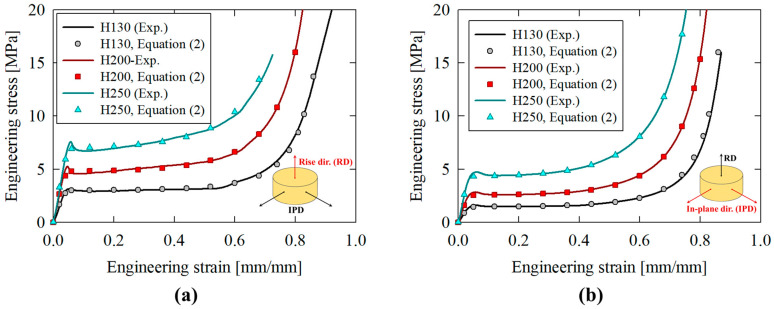
Quasi-static (10^−3^ s^−1^), compressive stress/strain responses for H130, H200 and H250 PVC foam in the (**a**) rise direction (RD) and (**b**) in-plane direction (IPD) and the corresponding theoretically predicted responses utilizing the Avalle constitutive model [62] from Equation (2).

**Figure 3 materials-16-06282-f003:**
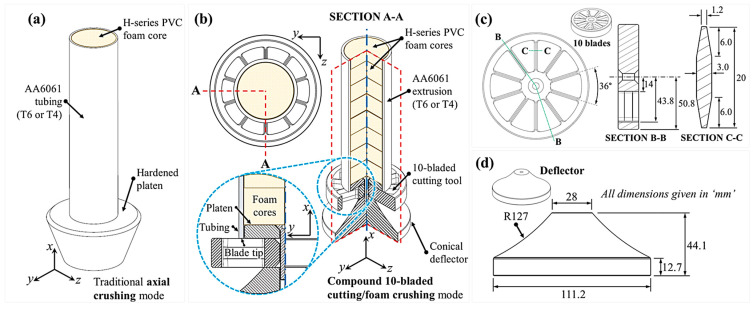
Schematics of the energy-absorbing apparatuses for AA6061/H-series PVC foam-filled extruded tubing, in T6 and T4 temper conditions, subjected to (**a**) axial crushing and (**b**) 10-bladed hybrid cutting/clamping with detailed views of the (**c**) 10-bladed cutter and (**d**) conical deflector.

**Figure 4 materials-16-06282-f004:**
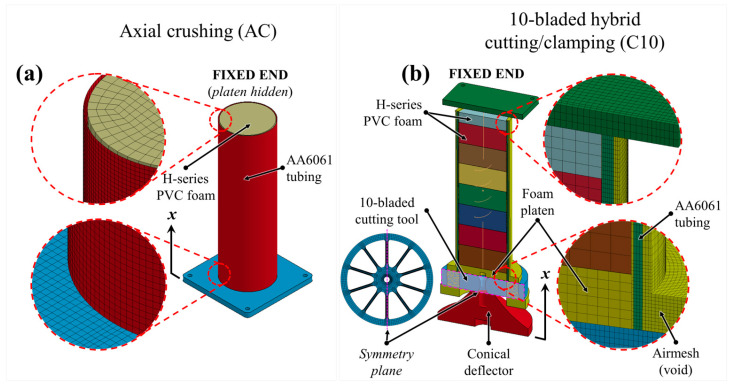
Finite element meshes of the PVC foam-filled AA6061 tubing subjected to (**a**) axial crushing, generated as a full model, and (**b**) 10-bladed hybrid cutting/clamping, generated as a half-symmetric model; the presence of the translational spring for the C10 deformation mode is explained in Section 2.4.2. The foam cores and complementary inputs were omitted for the hollow structures.

**Figure 5 materials-16-06282-f005:**
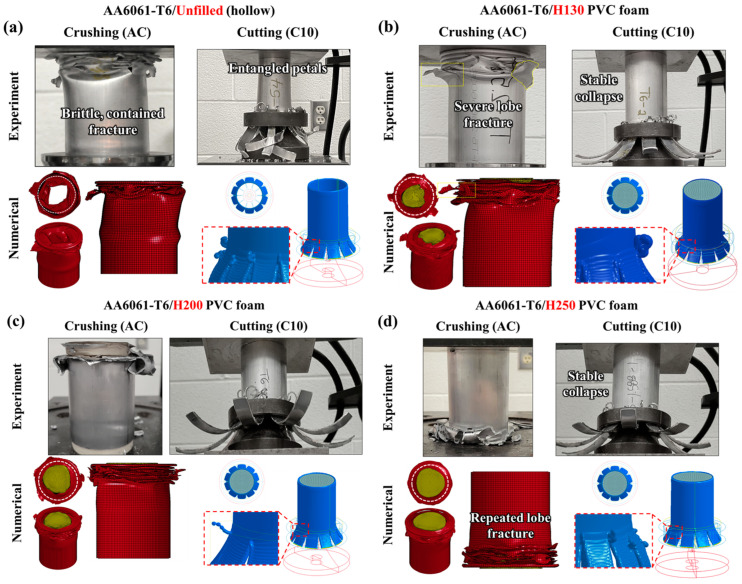
Observed and simulated deformation profiles for AA6061-T6 extruded tubing in (**a**) unfilled and (**b**) H130, (**c**) H200 and (**d**) H250 foam-filled configurations subjected to axial crushing and 10-bladed cutting; note the reduced stability of the crushing mode augmented with foam cores due to worsening lobe fracture compared to improved stability during cutting.

**Figure 6 materials-16-06282-f006:**
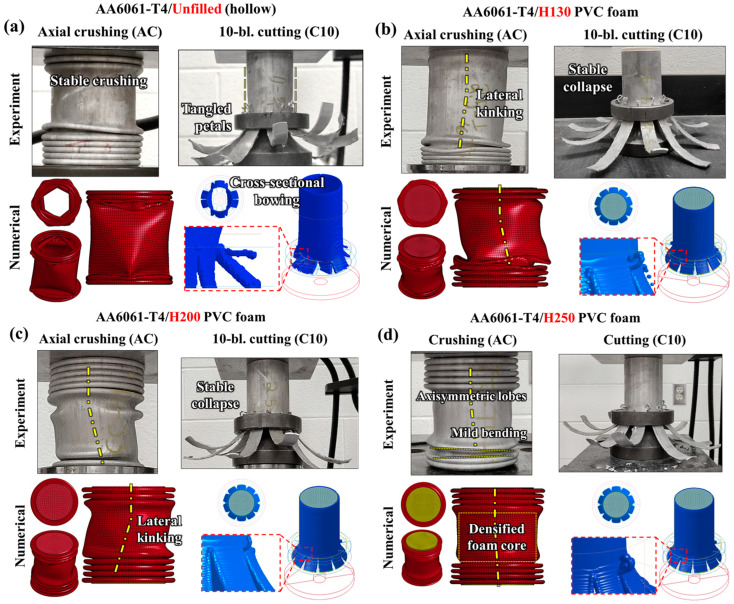
Observed and simulated deformation profiles for AA6061-T4 extruded tubing in (**a**) unfilled and (**b**) H130, (**c**) H200 and (**d**) H250 foam-filled configurations subjected to axial crushing and 10-bladed cutting, reduced stability and kinking due to densified foam core compaction for the crushing mode, in contrast with the annular support which prevented lateral bowing for cutting.

**Figure 7 materials-16-06282-f007:**
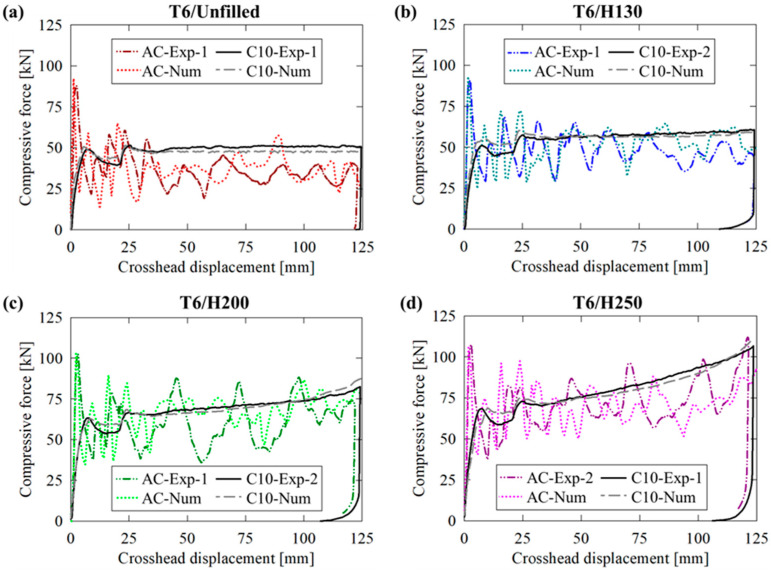
Experimentally recorded and numerically predicted force/displacement responses for AA6061-T6 extruded tubing in (**a**) unfilled and (**b**–**d**) H130, H200 and H250 foam-filled configurations, respectively, subjected to axial crushing (AC) and 10-bladed hybrid cutting/clamping (C10).

**Figure 8 materials-16-06282-f008:**
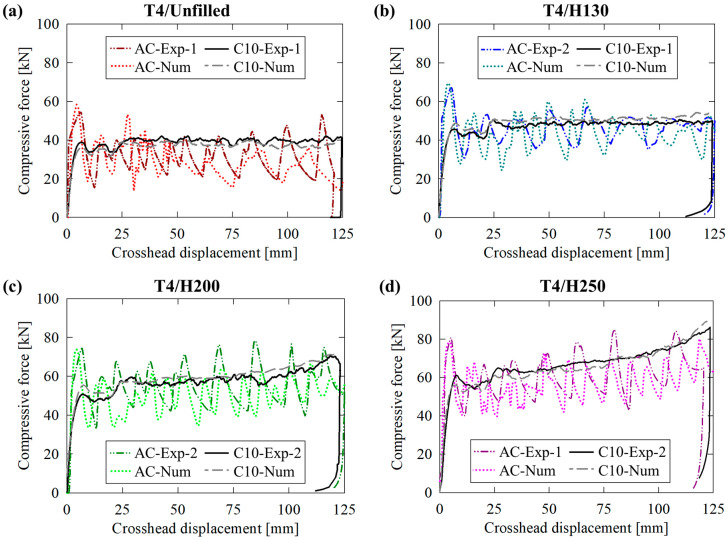
Experimentally recorded and numerically predicted force/displacement responses for AA6061-T4 extruded tubing in (**a**) unfilled and (**b**–**d**) H130, H200 and H250 foam-filled configurations, respectively, subjected to axial crushing (AC) and 10-bladed hybrid cutting/clamping (C10).

**Figure 9 materials-16-06282-f009:**
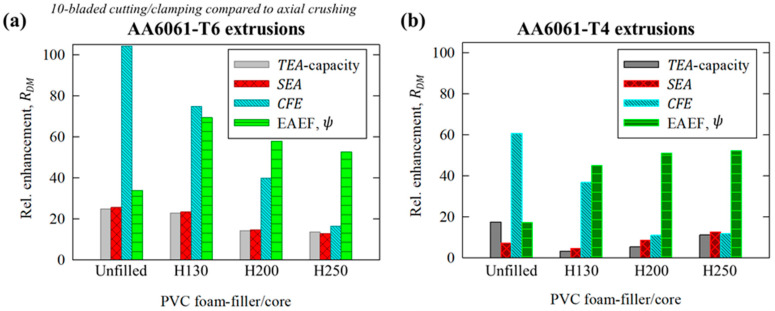
Relative enhancement, *R_DM_*, in energy-absorbing performance metrics for the 10-bladed cutting/clamping mode compared to axial crushing for AA6061 extruded tubing in (**a**) T6 and (**b**) T4 temper conditions, with and without PVC foam fillers.

**Figure 10 materials-16-06282-f010:**
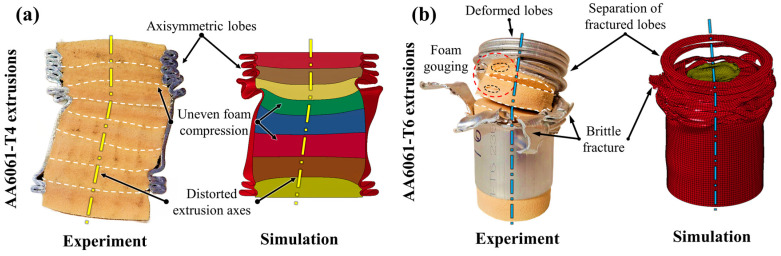
Experimentally and numerically observed principal deformation mechanisms in H130 foam-filled AA6061 extruded tubing in (**a**) T4 and (**b**) T6 temper conditions subjected to axial crushing; note the lateral kinking with uneven foam core compression for the former and widespread brittle lobe fracture for the latter configuration.

**Figure 11 materials-16-06282-f011:**
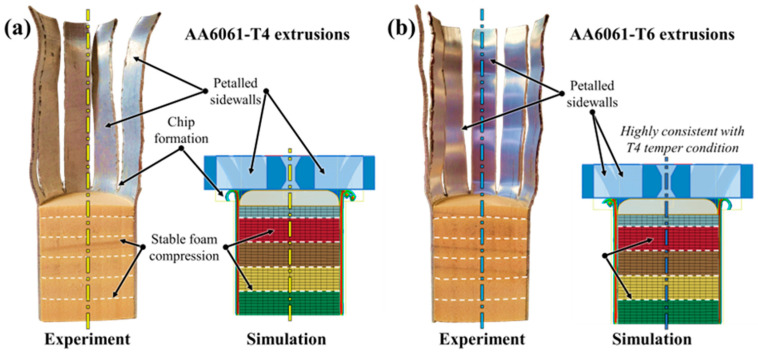
Experimentally and numerically observed principal deformation mechanisms in H130 foam-filled AA6061 extruded tubing in (**a**) T4 and (**b**) T6 temper conditions subjected to 10-bladed cutting/clamping, even foam compression and near-identical deformation profiles were observed despite the disparate properties of the as-received and annealed temper conditions.

**Figure 12 materials-16-06282-f012:**
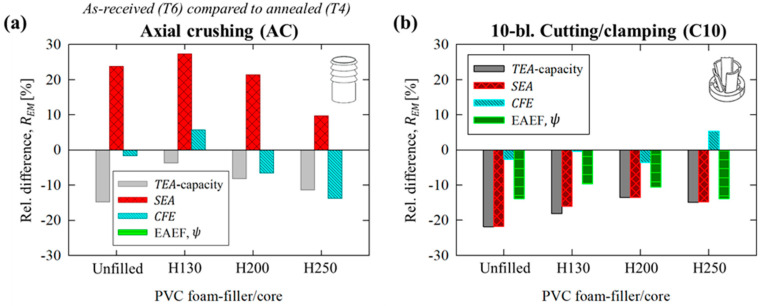
Relative difference in performance metrics for T4 (annealed) compared to T6 (as-received) tempered AA6061 energy absorbers, *R_EM_*, subjected to (**a**) axial crushing and (**b**) 10-bladed cutting/camping, with and without PVC foam fillers.

**Figure 13 materials-16-06282-f013:**
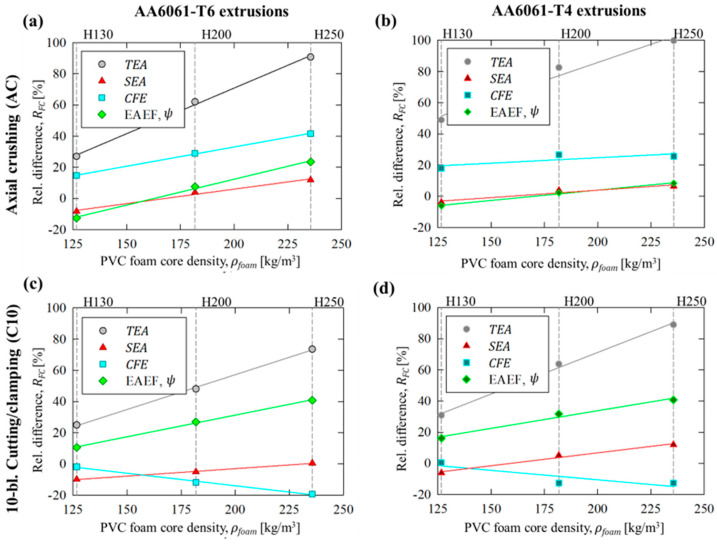
Relative difference in performance metrics for AA6061 extruded tubing in T6 and T4 temper conditions, respectively, subjected to (**a**,**b**) axial crushing and (**c**,**d**) 10-bladed cutting/clamping with increasingly dense PVC foam cores compared to the reference, unfilled cases.

**Figure 14 materials-16-06282-f014:**
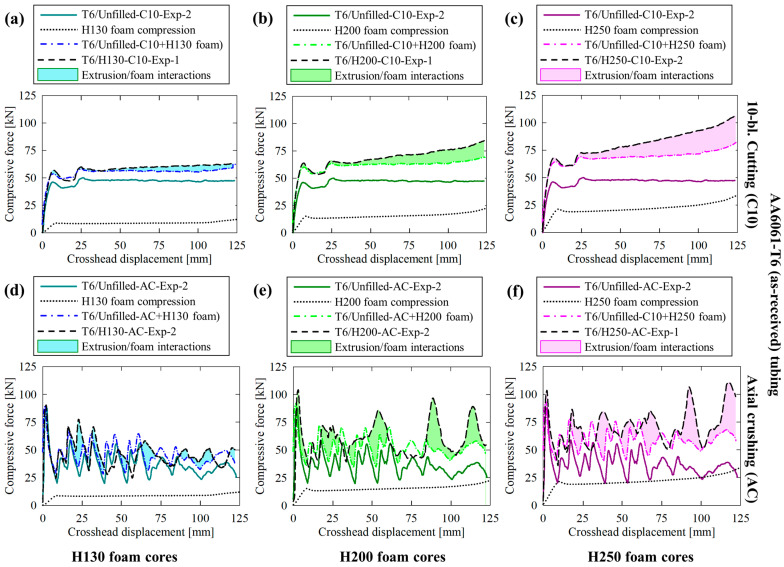
Comparisons between force responses for hollow AA6061-T6 (as-received) extruded tubing subjected to (**a**–**c**) 10-bladed cutting and (**d**–**f**) axial crushing with H130, H200 and H250 foam cores, respectively, with the tube/foam core interactions identified as the shaded areas, determined as the differences between the actual results for the composite structures and algebraic sums.

**Figure 15 materials-16-06282-f015:**
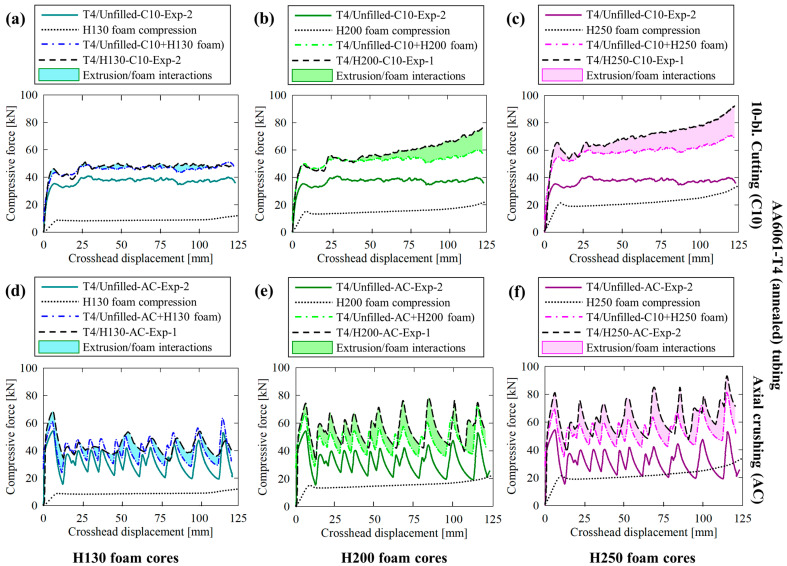
Comparisons between force responses for hollow AA6061-T4 (annealed) extruded tubing subjected to (**a**–**c**) 10-bladed cutting and (**d**–**f**) axial crushing with H130, H200 and H250 foam cores, respectively, with the tube/foam core interactions identified as the shaded areas, determined as the differences between the actual results for the composite structures and algebraic sums.

**Figure 16 materials-16-06282-f016:**
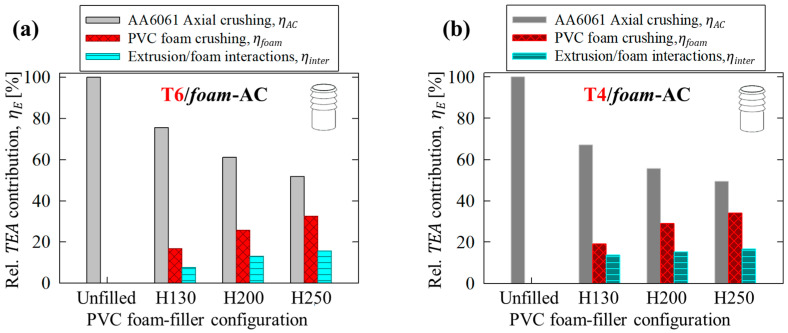
Average relative contributions, *η*, toward *TEA* capacity for each principal deformation mechanism in unfilled and PVC foam-filled AA6061 extruded tubing, in (**a**) T6 and (**b**) T4 temper conditions, subjected to axial crushing (AC).

**Figure 17 materials-16-06282-f017:**
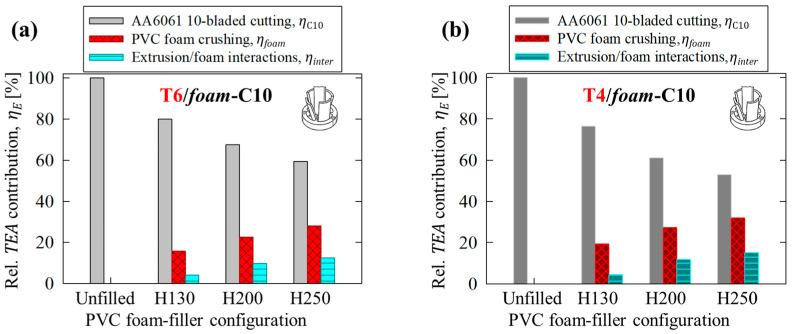
Average relative contributions, *η*, toward *TEA* capacity for each principal deformation mechanism in unfilled and PVC foam-filled AA6061 extruded tubing, in (**a**) T6 and (**b**) T4 temper conditions, subjected to 10-bladed cutting/clamping (C10).

**Figure 18 materials-16-06282-f018:**
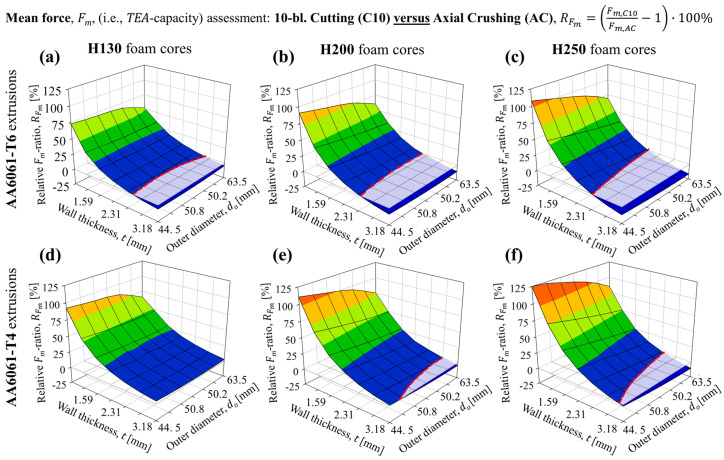
Contour plots of the mean force relative comparison ratio, $R_{F_{m}}$, for composite AA6061/PVC foam structures composed of (**a**–**c**) T6 and (**d**–**f**) T4 temper conditions with H130, H200 and H250 foam cores, respectively, subjected to axial crushing and 10-bladed cutting/clamping, with the latter mode taken as the reference case. Equivalent performance between crushing and cutting deformation modes, for a given tube/foam/geometric pairing, is denoted by the dashed lines.

**Figure 19 materials-16-06282-f019:**
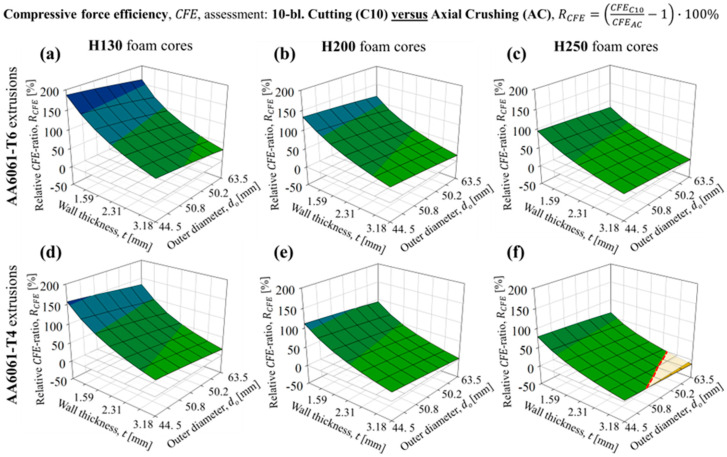
Contour plots of the compressive force efficiency comparison ratio, $R_{CFE}$, for composite AA6061/PVC foam structures composed of (**a**–**c**) T6 and (**d**–**f**) T4 temper conditions with H130, H200 and H250 foam cores, respectively, subjected to axial crushing and 10-bladed cutting/clamping, with the latter mode taken as the reference case.

**Figure 20 materials-16-06282-f020:**
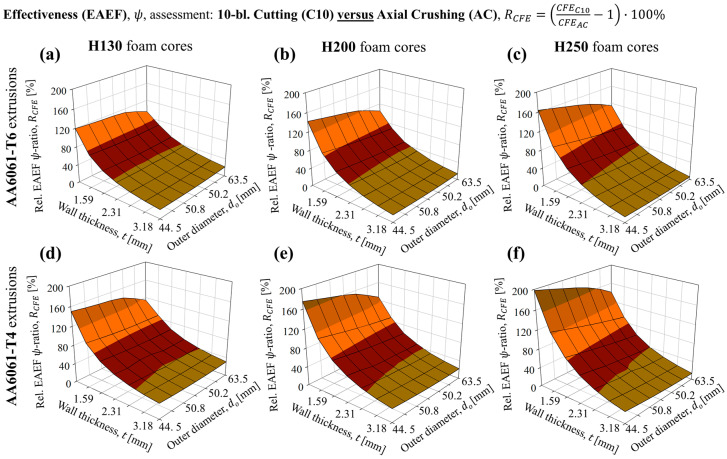
Contour plots of the energy-absorbing effectiveness comparison ratio, $R_{\psi }$, for composite AA6061/PVC foam structures composed of (**a**–**c**) T6 and (**d**–**f**) T4 temper conditions with H130, H200 and H250 foam cores, respectively, subjected to axial crushing and 10-bladed cutting/clamping, with the latter mode taken as the reference case.

**Figure 21 materials-16-06282-f021:**
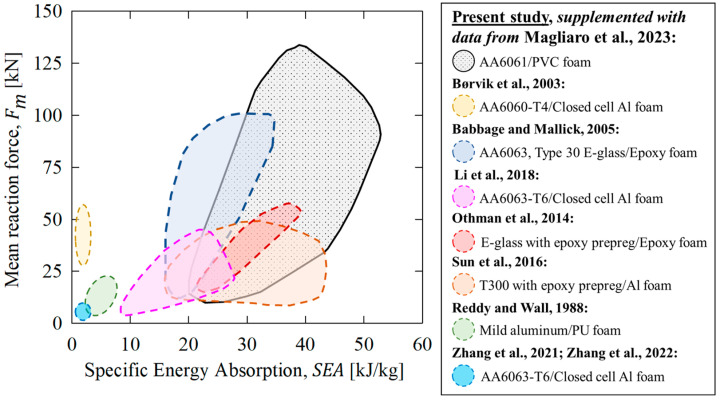
Ashby plot summarizing the range of potential mean forces, *F_m_*, with respect to specific energy absorption, SEA, achievable with the proposed 10-bladed cutting/foam crushing system (present study and [57]) compared to alternative sacrificial foam-filled energy absorbers [27,29,31,33,34,51,52,74].

**Table 1 materials-16-06282-t001:** Critical orthotropic mechanical properties and constitutive terms for the Avalle constitutive model (Equation (2)) for H-series PVC foams in the rise (RD) and in-plane (IPD) directions [56,63].

PVC Foam Material	H130		H200		H250	
Loading Plane	RD	IPD	RD	IPD	RD	IPD
Densification strain, *ε_d_* (mm/mm)	0.57	0.57	0.62	0.57	0.53	0.55
Plateau stress, *σ_P,foam_* (MPa)	2.8	N/A	4.9	N/A	7.2	N/A
Poisson’s ratio, *ν_foam_*	0.015	N/A	0.074	N/A	0.089	N/A
Elastic modulus, *E_foam_* (MPa)	85.5	44.5	132.2	80.0	168.0	131.3
Plateau stress, *σ_Y,foam_* (MPa)	2.7	N/A	4.7	N/A	6.8	N/A
Shear modulus, *G_foam_* (MPa)	56.7	56.7	87.6	87.6	111.4	111.4
Hardening coefficient, *K_h_* (MPa)	0.40	0.37	0.87	0.81	1.58	1.67
Plateau coefficient, *K_p_* (MPa)	2.91	1.45	4.79	2.55	6.92	4.34
Hardening exponent, *m*	−14.34	−21.81	−14.34	−21.81	−14.34	−21.81
Densification exponent, *n*	1.85	1.99	1.85	1.99	1.85	1.99

**Table 2 materials-16-06282-t002:** Testing scope for AA6061 extruded tubing, with and without H-series PVC foam cores, subjected to quasi-static axial crushing (AC) and 10-bladed hybrid cutting/clamping (C10).

Extruded Tubing Materials	Internal PVC Foam Cores	Outer Diameter, *d_o_* (mm)	Wall Thickness, *t* (mm)	Deformation Modes	Loading Rate
AA6061-T6, AA6061-T4	Unfilled, H130, H200, H250	63.5	1.588	AC, C10	10 mm/min

**Table 3 materials-16-06282-t003:** Johnson–Cook material model parameters for extruded AA6061 tubing [45,65].

Material	*A* (MPa)	*B* (MPa)	*c*	*n*	*E* (GPa)	*K* (GPa)	*G* (GPa)	ν
6061-T6 *	324	114	0.002	0.42	68.9	67.5	25.9	0.33
6061-T4	114	256	0.015	0.34	65.3	64.0	24.7	0.33

* A full Johnson–Cook model with damage parameters, *D*_1_ through *D*_5_ = {−0.77, 1.45, −0.47, 0.00, 1.60}, was implemented for the axial crushing models involving tubing in a T6 temper condition.

**Table 4 materials-16-06282-t004:** Parametric scope for theoretical study of AA6061 extruded tubing with H-series PVC foam cores subjected to axial crushing (AC) and 10-bladed compounded cutting/foam crushing (C10).

Def. Mode	Extruded Material	Outer Diameter, *d_o_* (mm)	Wall Thickness, *t* (mm)	PVC Foam-Filler Config.
{AC, C10}	AA6061-T6, AA6061-T4	{44.5, 47.6, 50.8, 54.0, 57.2, 60.3, 63.5}	{0.79, 1.19, 1.59, 1.98, 2.31, 2.79, 3.18}	H130, H200, H250

**Table 5 materials-16-06282-t005:** Average experimentally determined performance metrics for hollow and PVC foam-filled AA6061-T6 extruded tubing subjected to axial crushing (AC) and 10-bladed cutting (C10).

Material Pair	T6/Unfilled		T6/H130		T6/H200		T6/H250	
Def. Mode	AC	C10	AC	C10	AC	C10	AC	C10
*TEA* (kJ)	4.55	5.68	5.78	7.10	7.37	8.41	8.68	9.86
*SEA* (kJ/kg)	25.36	31.83	23.26	28.69	26.37	30.22	28.38	32.00
*F_m_* (kN)	36.90	46.04	46.97	56.34	60.29	67.89	70.12	79.51
*CFE* (%)	45.14	92.22	51.79	90.52	58.23	81.38	63.92	74.39
EAEF, *ψ*	1.83	2.45	1.60	2.71	1.97	3.11	2.26	3.45

**Table 6 materials-16-06282-t006:** Average experimentally determined performance metrics for hollow and PVC foam-filled AA6061-T4 extruded tubing subjected to axial crushing (AC) and 10-bladed cutting (C10).

Material Pair	T6/Unfilled		T6/H130		T6/H200		T6/H250	
Def. Mode	AC	C10	AC	C10	AC	C10	AC	C10
*TEA* (kJ)	3.78	4.44	5.63	5.81	6.90	7.27	7.55	8.39
*SEA* (kJ/kg)	23.00	24.64	22.11	23.14	23.87	25.90	24.49	27.58
*F_m_* (kN)	31.44	36.01	45.24	47.27	55.39	58.72	62.15	67.75
*CFE* (%)	55.84	89.74	65.91	90.15	70.69	78.43	70.14	78.35
EAEF, *ψ*	1.80	2.11	1.69	2.45	1.84	2.78	1.95	2.97

## Data Availability

The data contained within this manuscript are part of an ongoing study, and, at this time, data sharing is not applicable.

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
