# Peer review of "Influence of Extruded Tubing and Foam-Filler Material Pairing on the Energy Absorption of Composite AA6061/PVC Structures"

_materials, 2023, doi:10.3390/ma16186282_

Round 1
Reviewer 1 Report
Dear Authors,
The paper submitted for review has a very large scope of research and certainly required a lot of work. At the same time, due to this scope of research, the paper is quite "hard" to read. This applies to all work in general. For example, in the research methodology itself, it is necessary to find out how the research was carried out and on what apparatus. The material itself is also described cursorily. In turn, a big plus are interesting drawings showing the course of the experiment. I think that saves the job. However, the research descriptions as well as the research results are confusing. In my opinion, scientific language should be relatively simple and easy for outsiders to interpret.
Detailed comments below:
I admit that the exact name of the composite in the subject of the work used in the research is not necessary.
Next, is the word "extrusion" in the title correct? In general, extrusion is forcing the raw material through a small hole. This is probably not the case here.
Line 28-71: I have to admit that this introduction is quite hard to read. This is probably due to some disorder, try to edit it again and correct it.
Line 75: You need to better describe/emphasize the innovation of your research. Write why your research is relevant in light of the current scientific literature.
Line 78: Then you must formulate the scientific purpose of your research. This is necessary, and this is missing here.
Line 97: In my opinion, the methodology should be redrafted. You must describe exactly what constitutes the research material and what you are researching. Also describe the scientific equipment in detail. On the one hand, the division into small chapters was to facilitate navigation in the methodology, on the other hand, it only complicated this work.
Sort carefully what is the empirical part and what is the theoretical part.
Line 106: Each material or research apparatus should be accurately described, eg (name: manufacturer, city, country). Review and correct all methodology.
Line 844: The conclusions are extensive, but I have no comments on them.
Author Response
Please see the attached PDF file summarizing both Reviewers' comments and the authors responses.

Reviewer 2 Report
Comments can be found in the attached file.

Author Response
Please see the attached PDF file containing both Reviewers' comments and the authors' responses.

Round 2
Reviewer 1 Report
Dear Authors,
Thank you for considering my comments. I accept all replies and corrections. In my opinion, the article is good.